# Prevalence of Internet Addiction during the COVID-19 Outbreak and Its Risk Factors among Junior High School Students in Taiwan

**DOI:** 10.3390/ijerph17228547

**Published:** 2020-11-18

**Authors:** Min-Pei Lin

**Affiliations:** Department of Educational Psychology and Counseling, National Taiwan Normal University, Taipei 10610, Taiwan; lmmpp@ntnu.edu.tw

**Keywords:** adolescents, COVID-19, internet addiction, prevalence, risk factors

## Abstract

The coronavirus disease 2019 (COVID-19) outbreak has significantly disrupted normal activities globally. During this epidemic, people around the world were expected to encounter several mental health challenges. In particular, Internet addiction may become a serious issue among teens. Consequently, this study aimed to examine the prevalence of Internet addiction and identify the psychosocial risk factors during the COVID-19 outbreak. This study was constructed using a cross-sectional design with 1060 participants recruited from among junior high school students around Taiwan using stratified and cluster sampling methods. Taiwan’s first COVID-19 case was diagnosed on 28 January 2020. New cases exploded rapidly in February, and as a result, participants were surveyed during March 2 through 27 March 2020. The prevalence of Internet addiction was found to be 24.4% during this period. High impulsivity, high virtual social support, older in age, low subjective well-being, low family function, and high alexithymia was all independently predictive in the forward logistic regression analyses. The prevalence rate of Internet addiction was high among junior high school students during the COVID-19 outbreak. Results from this study can be used to help mental health organizations and educational agencies design programs that will help prevent Internet addiction in adolescents during the COVID-19 pandemic.

## 1. Introduction

Globally, the coronavirus disease 2019 (COVID-19) pandemic has significantly disrupted normal activities of daily life [1]. Currently there is no effective prescription drug supported by reliable evidence that can treat the severe acute respiratory syndrome coronavirus [2]. As of April 2020, approximately 3 billion people worldwide were required to stay-at-home, and more than 130 countries have ordered some level of restrictions to limit movement in order to prevent the spread of COVID-19 [3]. Likewise, stay-at-home quarantines and mandates have escalated the consumption of digital entertainment [1]. During this crisis, mental health challenges ranging from sleep disturbances, anxiety, phobia, panic to dissociative like symptoms are expected to increase [4]. Addictive behaviors could arise potential problems during lock-downs and consequently, additional behavioral addictions may arise pertaining towards the teenage population [5]. Previous research on the impacts of disasters such as the severe acute respiratory syndrome epidemic, terrorist incidents, natural disasters and have shown growth rates of addictive behaviors, including smoking, excessive alcohol use, and Internet addiction (IA) [6,7]. In comparison with adults, addiction vulnerability appears to be a more problematic issue among adolescence [8]. There is a natural tendency for this age group to use the Internet and thus easily develop IA behavior [9]. Adolescents typically have flexible living schedules, unlimited Internet access, and freedom from parental interference [10]. Nearly 90% of students are physically cut off from their schools due to the COVID-19 pandemic, and technology has become necessary to enable students to access educational materials, to interact with each other, and to do what students need to do most: play [3,11]. Therefore, it is important to understand how the COVID-19 pandemic has influenced the use of technology during this unique period, and whether there is a change in IA prevalence. Therefore, it is important to identify the psychosocial risk factors related to IA of adolescents during the COVID-19 pandemic. Moreover, gaining a full understanding of the developmental process of IA is increasingly vital so that researchers, student affairs professionals and mental health organizations can discover possible underlying mechanisms, and associations between exposure outcomes [12].

Many previous studies have focused on adolescents and examined the associations between psychosocial risk factors and IA [13]. Comprehensively, this study applied the theory of triadic influence (TTI) [14,15] in attempt to understand the psychosocial risk factors for IA in adolescents during the COVID-19 outbreak. The TTI integrates elements of various theories on health behavior and categorizes these behaviors into three fundamental domains of influence (cultural/attitudinal, social/interpersonal, and intrapersonal) [16]. Recently, this theory has been used as a foundation to inspect addictive behaviors, including alcohol drinking, smoking, and IA [16,17,18]. The aim of this study is not to examine a theoretical model, but rather to organize statistically significant associates as a means of elucidating the multiple etiological nature of IA, in which this theory gives an appropriate theoretical guide. Based on the quarantines of the COVID-19 pandemic and stay-at-home mandates, this study focused on the intrapersonal and social stream of influence. Intrapersonal factors can include emotional instability, impulsivity, depressed mood, temperamental personalities, and low self-esteem [15]. Social factors can include lack of parental support, strong desires to and strong attachment to pleasure peers, weak desires to and weak attachment to pleasure family members, home strain, and negative evaluations from parents [15].

On the other hand, past researches have shown that higher levels of neuroticism [19,20], greater impulsivity [20,21], higher levels of depression [22,23], alexithymia [24,25], self-esteem [26,27], and subjective well-being [28,29] are all associated with IA. Furthermore, previous research also indicated sociological factors, including actual social support [30,31], virtual social support [32,33], and family function [34,35] to be related to IA.

In spite of the numerous psychological and sociological risk factors that were found to affect IA, few studies have examined the importance of psychological and sociological risk factors pertaining to junior high school students. Furthermore, past studies have investigated a minimal number of psychological and/or sociological risk factors, in which may restrict the comprehensiveness of understanding the relationship. Having limited resources ad time, governmental agencies and practices often choose only up to three to five important psychosocial risk factors to focus during each preventive implementation. This is also deemed true during the COVID-19 outbreak. For this reason, integration of three to five related psychosocial risk factors in IA is needed. Most importantly, to our knowledge, this is the first study to examine the prevalence rate of IA during the COVID-19 outbreak and its psychosocial risk factors of adolescents. Therefore, this study aimed to inspect the prevalence rate of Internet addiction and the associations of psychological/sociological risk factors and Internet addiction in Taiwan. This study hypothesized that neuroticism, impulsivity, depression, alexithymia, and virtual social support were positively associated with IA, and self-esteem, subjective well-being, actual social support, and family function were negatively related to IA.

## 2. Materials and Methods

### 2.1. Participants and Procedures

The present study was constructed with a cross-sectional design during the COVID-19 outbreak in Taiwan. Taiwan’s first COVID-19 case was diagnosed on 28 January 2020. New cases exploded rapidly in February, and as a result, schools in Taiwan extended their winter breaks by two weeks, including junior high schools, which reopened schools on February 25. The present study recruited 1244 junior high school students from three junior high schools located in northern Taiwan (Taipei City, New Taipei City, and Taoyuan City) using stratified and cluster (by class) sampling methods. Participants were surveyed during 2 March through 27 March 2020. Excluding those who did not participate in the administration of the surveys and invalid questionnaires, a total of 1060 valid surveys were completed (*Mage* = 14.66, *SD* = 0.86 years), corresponding to a response rate of 85.21%.

The present study protocol was reviewed and approved by the institutional review board of the corresponding author’s affiliated institution. Prior to implementing the assessments, this study obtained written informed consent from the school principals or guidance director, guidance counselor, and the teachers of the administered classes. A pre-training session regarding the procedure of the administration was given to the guidance counselors by the research team. Guidance counselors were instructed to administer the surveys in groups during class time. During the administration, the goal, procedure, research ethics and confidentiality was stressed to gain truthful responses from the participants. This study also obtained written informed consent from the participants’ guardians as well as the participants. Data analysis was conducted only on those who provided both sets of informed consent. After the completion of the surveys, students were notified that the results of their responses will be individually feed-backed to them during the following semester.

### 2.2. Measures

#### 2.2.1. The Socio-Demographic Measures

Gender (female = 1 and male = 2), age, and marital status of parents were assessed.

#### 2.2.2. The Chen Internet Addiction Scale (CIAS)

The CIAS is a 26-item self-reported questionnaire [36]. Modifying the diagnostic criteria of pathological gambling and substance dependence in Diagnostic and Statistical Manual of Mental Disorders-Fourth Edition-Text Revision [37], Ko et al. [38] developed the Diagnostic Criteria for IA, which comprised of 9 IA symptoms. The cut-off of six out of the nine provided the best diagnostic accuracy [38]. In a study that recruited 468 high school students, Ko et al. [39] asked them to complete both diagnostic interview and the CIAS, and discovered a CIAS > 63 cut-off point was the optimal diagnostic cutoff to differentiate between IA and non-IA. The diagnostic cutoff constitutes for high diagnostic accuracy (87.6%), Cohen Kappa (0.61), diagnostic odds ratio (26.17), and specificity (92.6%) to discriminate cases with IA from that of non-cases among high school students [39]. The discriminative aspects of this scale allows it to be a dependable diagnostic tool to be used in a one-stage epidemiological study [39]. Students with CIAS > 63 scores were categorized as IA diagnosis group.

#### 2.2.3. Shortened Chinese Version of Five-Factor Inventory—Neuroticism Subscale

This study used the neuroticism subscale of the Shortened Chinese version of Five-Factor Inventory to assess neuroticism personality traits [40,41]. This inventory selected from the NEO Five-Factor Inventory has 31 items [40] and displayed acceptable validity and reliability of Chinese adolescents [41]. In the current study, the neuroticism subscale includes six items and also displayed good Cronbach’s alpha coefficient (α = 81)

#### 2.2.4. The Barratt Impulsivity Scale—Short-Form

This scale contains 10 items and has satisfactory construct validity [42]. The confirmatory factor analysis showed non-planning subscale and motor impulsiveness subscale [43]. The scale showed acceptable internal consistency (α = 70) in this study.

#### 2.2.5. Depression Anxiety Stress Scale (DASS)—Depression Subscale

The depression subscale of the Chinese version of the DASS was used in the current study [44]. The depression subscale has seven items, which included items such as “I feel that life was meaningless”. The DASS demonstrated satisfactory internal consistency and factor structure among the Chinese population [44]. The depression subscale had a Cronbach’s α of 0.85 in the present study.

#### 2.2.6. The Rosenberg Self-Esteem Scale

Global self-esteem was assessed using a ten item self-reported scale on a 6-point Likert scale. This scale has good construct validity and reliability [45]. The scale also showed good internal consistency (α = 89) in the current study.

#### 2.2.7. The Toronto Alexithymia Scale-20

The alexithymia assessment is a self-report scale with twenty items and rated on a 5-point Likert scale. The scale has good factorial validity and test-retest reliability [46,47]. The scale also displayed good Cronbach’s alpha coefficient (α = 80) in this study.

#### 2.2.8. Chinese Happiness Inventory

This scale contains 10 items that evaluates subjective well-being and displayed good validity and reliability among adolescents [48]. This scale was rate on a 4-point Likert scale, and showed good internal consistency (α = 90) in the present study.

#### 2.2.9. Social Support Scale (SSS)

The SSS measures the perceived social support of friends and parents [49] and has 16-items rated on a 4-point scale. The scale was demonstrated to have good validity and reliability among adolescents in Taiwan [49]. The scale displayed good internal consistency (α = 90) in this study.

#### 2.2.10. Virtual Social Support Scale (VSSS)

Revised from the SSS [49], the VSSS assesses perceived social support from Internet acquaintances as distinguished from people we know in real life. Yeh et al. [49] incorporated two items to VSSS in order to fit the Internet environment, yielding 10 items to the final scale. The scale also showed good internal consistency (α = 95) in the present study.

#### 2.2.11. Brief Family Function Questionnaire

Family function was assessed using the Brief Family Function Questionnaire [50]. This questionnaire is a 22-item self-reported questionnaire with a 5-point Likert scale, which includes eight aspects of family functioning, including family conflict, family cohesion, affective responsiveness, affective involvement, communication, independence, problem solving, and role responsibility. This questionnaire demonstrated good validity and reliability among adolescents in Taiwan [51]. The emotional involvement and conflict subscales were reverse scored, and higher scores indicated healthier family functioning. In the current study, the Cronbach’s alpha for the total scale was 92.

### 2.3. Statistical Analysis

First, descriptive analyses examining the IA prevalence rate was calculated. Then, the association between psychosocial risk factors and IA were inspected using Pearson correlation, *t*-test, and χ^2^ test. Finally, in order to examine the association of IA among junior high school students, significant risk factors were selected and incorporated in the forward logistic regression analyses. SPSS version 18.0 for Windows was used for all analyses and the significance level was set at 05.

## 3. Results

IA was diagnosed based on the total score of the CIAS. Participants who scored higher than 64 and more on the CIAS were considered Internet addicted group among junior high school student [39]. Accordingly, the cutoff was utilized to examine the estimated prevalence rate of IA and identify the case group. The prevalence rate of IA was 24.4% among junior high school students in Taiwan during the COVID-19 outbreak (95% confidence interval, 21.8–27.0%).

Table 1 reveals that, except for gender and marital status of parents, age and all psychosocial risk factors were significantly different in the non-IA and IA groups. When compared to the non-IA group, the IA group was found to be significantly older in age, and have higher neuroticism, higher impulsivity, higher depression, higher alexithymia, lower self-esteem, lower subjective well-being, lower actual social support, higher virtual social support, and lower family function. In addition, Table 2 shows the bivariate correlations of study variables.

The results in Table 3 showed variables that significantly increased IA risk: high impulsivity, high virtual social support, high age, low subjective well-being, low family function, and high alexithymia. However, in the logistic regression model, neuroticism, depression, actual social support, and self-esteem were not associated with IA. Each factor was added separately in the logistic regression analysis, and discovered actual social support (Wald. coeff = 18.044, *p* < 0.001), neuroticism (Wald. coeff. = 70.198, *p* < 0.001), depression (Wald. coeff. = 62.807, *p* < 0.001), and self-esteem (Wald. coeff. = 55.832, *p* < 0.001) significantly and independently predicted IA, but did not significantly predict IA when additional psychological and sociological risk factors were entered into the forward logistic regression analyses.

## 4. Discussion

This study sought to investigate the prevalence rate of IA and related psychosocial risk factors among junior high school students in Taiwan during the COVID-19 outbreak. The CIAS cutoff score of 63/64 [39] indicated a 24.4% (95% confidence interval, 21.8–27.0%) IA prevalence in the examined sample. The prevalence rate appears to be higher than those found in previous researches regarding middle school student samples, including Kawabe et al.’s [52] Japan junior high school student sample (2.0%; aged 12–15 years), Stavropoulos, Alexandraki, & Motti-Stefanidi’s [53] Greek high school student sample (3.1%; mean age 16), Černja et al.’s [54] Croatia high school student sample (3.4%; aged between 15 and 20), Liu et al.’s [55] USA high school student sample (4.0%; age range = 14–18 years), Kuss et al.’s [56] Netherlands adolescents (3.7%; Mage = 14.2, SD = 1.1 years), Park et al.’s [57] South Korean middle and high school student sample (10.7%), Sasmaz et al.’s [58] high school student sample in Turkey (15.1%; mean age was 16.1 ± 0.9 years), Lau et al.’s [59] Hong Kong secondary school student sample (16.0%; mean age = 14.53 years), Tan et al.’s [60] China junior high school student sample (17.2%), Lin et al.’s [13] Taiwan senior high school student sample (17.4%; Mage = 15.83, SD = 0.38 years), and Di Nicola et al.’s [61] Italy high school student sample (22.1%; mean age was 16.47 ± 4.85 years). The prevalence rate of IA appears to vary within a wide range of middle school students, however, the prevalence rate of IA in this study was observed to be the highest. Past research incorporated both junior and senior high school students, which may increase the prevalence due to the higher in age average. However, this study solely focused on junior high school students, with only a mean age of 14.66 years, but the results showed this group to have the highest IA prevalence. There is a probability that having a youngest age with the highest IA prevalence may be associated with the COVID-19 outbreak. Taiwan’s first COVID-19 case was diagnosed on 28 January 2020. New cases exploded rapidly in February, and as a result, schools in Taiwan extended their winter breaks by two weeks, and schools reopened on 25 February. Participants were surveyed during 2 March through 27 March 2020. In other words, junior high school students not only experienced a prolonged winter vacation, but may also encounter various psychological impacts and stress due to the COVID-19 outbreak. Such an experience makes it easier to develop a dependence on the Internet, and be more likely to become IA. Such a high prevalence deserves attention by mental health organization and educational agencies to construct intervention strategies. After all, the average age of the studied group was only 14.66 years old, and the COVID-19 pandemic is still in progress effecting various aspects of daily life.

In the psychological dimension, this study discovered that impulsivity was positively related to IA, which was consistent with past researches [13,20,21,62]. Potential contributions of psychosocial risk factors of IA were further analyzed using a forward logistic regression analyses. This study found that impulsivity had the greatest influence on IA. A possible explanation may be that individuals are more likely to overlook the long-term benefits when confronted with instant gratifications on the Internet, increasing the possibility of IA development [13]. Li et al. [62] used delay and probability discounting tasks to examine such a hypothesis and revealed that, irrespective of the reward signal and financial amount, those who were Internet addicted were more likely to disregard delayed amounts more abruptly than the non-IA adolescents. Consequently, when junior high school students with high impulsivity use the Internet during the COVID-19 outbreak, their inattentiveness to postpone positive outcomes proliferations the chance of developing Internet addiction. As a result, this study suggest that mental health and educational organizations design programs focus on educating junior high school students to improve skills in building behavior plans and enhance their attempts in delayed gratification in preventing to development of IA during the COVID-19 pandemic.

This study also revealed that IA is negatively and significantly related to subjective well-being and positively and significantly associated with alexithymia in the psychological dimension, which corresponds with the results of previous studies on IA [24,25,28,29]. Subjective well-being is defined as a person’s cognitive and emotional evaluations of their life, including peace, happiness, life satisfaction, and fulfillment [63]. On the other hand, alexithymia suggests difficulty in managing emotional expression or a deficit in emotion regulation [64]. Adolescents who show high levels of alexithymia tend to experience more emotional pains [65]. In other words, when junior high school students encounter the COVID-19 outbreak, they are faced with various psychological stress and impact, delayed going to school by two weeks, and having families, schools and the entire society needing to confront enormous stress throughout the entire February and March. Junior high school students who feel a lack of happiness, peace, fulfillment, and life satisfaction, or is unable to be aware or express their emotions and in turn experience more emotional pains, may be more likely to vent their emotions through the Internet as a source for them to gain happiness, peace, fulfillment, and life satisfaction. This experience may increase the risk of becoming IA. Thus, it becomes important to pay attention to the psychological aspects, including subjective well-being, emotional awareness and emotional expression of junior high school students during the COVID-19 pandemic period.

In the sociological dimension, consistent with previous studies, this study also discovered that high virtual social support and low family function independently contributed in the prediction of IA, respectively [32,33,34,35]. Participation in virtual social network activities may allow individuals to form close social relationships with online friends, and these relationships might provide them a sense of understanding, allowing them to feel secure and know that there is someone online to listen to them when needed [32]. There is a high possibility that being sheltered at home, junior high school students may increase their Internet use in order to gain related COVID-19 information, and as a result, students may feel that they can count on their online friends to understand them when needed. However, repeatedly make unsuccessful efforts to limit or stop their use may accelerate the development of IA. In contrast, low family function increases the risk of IA during the COVID-19 outbreak. Family functions in an important indicator of how family members interact with each other and how the strengths and weaknesses of family structure [66]. During the COVID-19 pandemic, everyone is bombarded with great stress, including junior high school students and their family members. Having a poor family function at the time being will cause an increase likelihood for junior high school students to increase their use the Internet in attempt to try and alleviate the stress caused by their family. This is especially the case when everyone is forced to shelter at home. In other words, IA may be a behavioral symptom that reflects poor family function among junior high school students during the pandemic. Consequently, in the sociological dimension, special attention should be placed on junior high school students during the COVID-19 pandemic and monitor whether they become overly dependent of virtual social support in order to overpass the stress caused by the COVID-19 pandemic. Poor family function should also be noted due to its effect in generating an increased risk in IA risk as a result of students attempting to relinquish stress caused from family pressure.

The present study found that depression and neuroticism were significantly lower, and actual social support and self-esteem were significantly higher with non-IA group compared to IA group. These results corresponded with previous studies on IA [19,20,22,23,26,27,30,31]. However, when entered after other psychological and sociological risk factors, these factors could not significantly predict Internet addiction in the logistic regression analyses. Moreover, this study also added each factor separately in the logistic regression analysis, and discovered actual social support, neuroticism, depression, and self-esteem were able to significantly predict IA, but did not predict IA when other psychological and sociological risk variables were included into the regression model. There may be possible mediation effects caused by other psychosocial risk factors in the relationship. Comprehensively, this study incorporated multiple psychological and sociological risk factors correlated with IA during the COVID-19 outbreak.

Interpretations of our findings should be taken with caution and keep in mind of the following limitations. First, the research was based on a cross-sectional design, and thus we cannot eliminate the possibility that IA may already be present among the 24.4% Taiwanese junior high school students (IA-group) prior to COVID-19 outbreak. Further studies should conduct follow-up studies in order to clarify whether the rise in prevalence was due to the COVID-19 outbreak. Second, solely recruiting the participation of three junior high schools limited the generalizability of study results, and outcomes may not be appropriate to all junior high school students with Internet addicted problems, especially those who were required to stay at home due to the pandemic or were addicted to the Internet beforehand. Furthermore, the three junior high schools that participated in the study were municipality, each located in Taipei City, New Taipei City, and Taoyuan City, respectively, which have similar socioeconomic features and may also limit the generalizability of study results. Third, students from families of high risk socioeconomic factors may be the ones more likely to increase their Internet use after the COVID-19 outbreak began, however, the present study did not include related variables. Thus, family background factors such as socioeconomic status, and area of residence should be taken into account in further studies. Lastly, all data gathered was obtained through self-reports, which heavily relied on a clear comprehension of the questions and honest reports, and can be effected by response bias. Further studies should use multiple-method assessments in order to provide a more comprehensive understanding of IA.

## 5. Conclusions

The COVID-19 pandemic has drastically disrupted the daily life of people around the world, while stay-at-home orders and quarantines have heightened the use of digital entertainment. Addictive behaviors could become a problem during the locked-down period and subsequently, Internet addiction could emerge as an important issue for adolescents. To our knowledge, this was the first study to investigate the prevalence rate of Internet addiction and examined the associations between Internet addiction and a number of psychological and sociological risk factors among junior high school students during the COVID-19 outbreak. This study showed that Internet addiction is fairly prevalent among junior high school students during the COVID-19 pandemic. High impulsivity, high virtual social support, older in age, low subjective well-being, low family function, and high alexithymia was all independently predictive in the forward logistic regression analyses in predicting Internet addiction. These results can serve a guideline for educational agencies and mental health organizations to design programs and policies that will help prevent Internet addiction during the COVID-19 pandemic. However, caution should be taken when interpreting the results. Although results are found during the alarmed state by COVID-19, it cannot be assured that they were different before the pandemic, and further studies should compare the results with that of other cultures and at different times.

## Figures and Tables

**Table 1 ijerph-17-08547-t001:** Socio-demographic and psychosocial risk factors between the non-IA and IA groups.

Risk Factor	Internet Addiction		χ^2^ or *t*	Effect Size
	Yes	No		
	(*n* = 254) *n* (%) or	(*n* = 788) *n* (%) or		
	Mean (SD)	Mean (SD)		
Socio-demographic factors				
Age	14.83(0.96)	14.61(0.82)	−3.39 **	0.25
Gender				
Female	125(49.02%)	379(47.91%)	0.09	0.01
Male	130(50.98%)	412(52.09%)		
Marital status of parents				
Marriage	185(74.90%)	613(80.34%)	3.33	0.06
Divorce	62(25.10%)	150(19.66%)		
Psychological factor				
Neuroticism	18.83(5.29)	15.45(5.20)	−9.02 ***	0.64
Impulsivity	33.61(7.50)	28.35(6.40)	−10.07 ***	0.75
Depression	6.41(4.55)	3.89(3.88)	−7.96 ***	0.59
Alexithymia	59.88(9.56)	52.51(10.77)	−10.38 ***	0.72
Self-esteem	33.17(8.77)	38.87(10.40)	8.64 ***	0.59
SWB	21.45(5.78)	25.30(6.16)	8.81 ***	0.64
Sociological factor				
Actual social support	46.56(8.17)	49.18(8.45)	4.33 ***	0.31
Virtual social support	22.53(8.40)	18.74(7.85)	-6.57 ***	0.47
Family function	70.58(15.27)	79.05(15.22)	7.75 ***	0.56

** *p* < 0.01; *** *p* < 0.001. Note: IA group is CIAS > 63; SWB = subjective well-being.

**Table 2 ijerph-17-08547-t002:** Bivariate correlations of study variables.

Variable.	1	2	3	4	5	6	7	8	9	10
1. Age	—									
2. Neuroticism	0.07 *	—								
3. Impulsivity	0.11 **	0.42 ***	—							
4. Depression	0.02	0.67 ***	0.37 ***	—						
5. Alexithymia	0.07 *	0.60 ***	0.44 ***	0.54 ***	—					
6. Self-esteem	−0.05	−0.62 ***	−0.27 ***	−0.65 ***	−0.58 ***	—				
7. SWB	−0.06	−0.56 ***	−0.24 ***	−0.63 ***	−0.54 ***	0.72 ***	—			
8. Actual social support	0.02	−0.16 ***	−0.07 *	−0.35 ***	−0.24 ***	0.40 ***	0.51 ***	—		
9. Virtual social support	−0.01	0.10 **	0.08 *	0.13 ***	0.09 **	−0.08 *	−0.05	0.15 ***	—	
10. Family function	−0.03	−0.29 ***	−0.17 ***	−0.41 ***	−0.34 ***	0.46 ***	0.54 ***	0.64 ***	−0.06	—
*M*	14.66	16.26	29.60	4.50	54.29	37.49	24.36	48.50	19.67	76.93
*SD*	0.86	5.41	7.05	4.19	10.95	10.33	6.29	8.49	8.15	15.70

* *p* < 0.05; ** *p* < 0.01; *** *p*< 0.001. Note: SWB = subjective well-being.

**Table 3 ijerph-17-08547-t003:** Forward logistic regression analyses in predicting IA from socio-demographic and psychosocial risk factors.

Psychosocial Risk Factor	Wald χ^2^	Odd Ratios	95% Confience Interval
Impulsivity	46.470 ***	1.091	1.064–1.119
Virtual social support	31.310 ***	1.059	1.038–1.081
Age	9.425 **	1.328	1.108–1.592
Subjective well-being	9.127 **	0.948	0.915–0.981
Family function	8.821 **	0.982	0.970–0.994
Alexithymia	5.385 *	1.023	1.003–1.042

* *p* < 0.05; ** *p* < 0.01; *** *p* < 0.001.

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
