# Peer review of "Prevalence of Internet Addiction during the COVID-19 Outbreak and Its Risk Factors among Junior High School Students in Taiwan"

_ijerph, 2020, doi:10.3390/ijerph17228547_

Round 1

Reviewer 1 Report

Reviewer comments

General

This is an interesting study on internet addiction during Covid-19 outbreak. Study comprises of junior high school students from three Taiwanese junior high schools.

Main weakness of this study is that as it is on statistical associations only, one cannot assess the causality or even try to do that. There is no proof that IA would have been already at level of 24.4 % among Taiwanese junior high school students prior to Covid-19 outbreak (even though this is quite plausible that the IA has increased due to Covid-19 outbreak). There is plenty of studies suggesting that the cause for the psychological symptoms (depression, neuroticism, self esteem, alexithymia etc.) examined, like in this study are also due to socioeconomic factors and the family background, like age, gender, family social status, family income, marital status of parents etc. Some references below. None of these were taken into account in this study. It could well be that the students from families of high risk socioeconomic factors were the ones who started to use more internet after Covid-19 outbreak began. That would indicate that IA is more of a confounder for the true reasons of depression, neuroticism etc. It is impossible to assess the study as none of these, commonly recorded factors of socioeconomic nature, were not taken into account. The least, other variables should be adjusted for these to account for any biases. Surely and hopefully, some basic demographics (gender, area of residence etc.) were collected.

Major comments

    • l. 30 update numbers to also give the most recent numbers (even though the study was during early outbreak)
    • t-test is for continuous variables and their means, your variables have only about 4-6 classes. E.g. chi-square might be more suited here.
    • Study and sampling design has not been described at all. In line 94, a sentence saying that “stratified and cluster sampling” was done, is not enough. There should be detailed description on where, how, with whom the sampling was done, ethical statements etc. Was any demographics collected, hopefully so, as it would save this study.
    • If in Taiwan, senior high school students have IA of 17.4 %, the 24.4 % among junior high school students obtained in this study is not much higher.
    • Any difference between schools? Were schools from similar socioeconomic areas?
    • You give no alternative for junior high school students what to do during Covid-19 lockdowns. If internet is forbidden, then what? Going out with friends is not an option. I would see this far more positively, or at least it cannot be concluded that the side effects were negative. It could also be that the use of IA helped those students with these mental health issues to cope better. This could be possibly analyzed with stratified methods.

DOI: 10.1371/journal.pone.0137506

DOI: 10.1186/s12889-018-5990-8

doi: 10.1016/j.jad.2020.08.026.

DOI: 10.1016/S0140-6736(13)62116-9

DOI: 10.1016/S0140-6736(11)60871-4

Author Response

In response to the comments from reviewer 1:

General

This is an interesting study on internet addiction during Covid-19 outbreak. Study comprises of junior high school students from three Taiwanese junior high schools.

Ans: Thank you for your positive affirmation!

Main weakness of this study is that as it is on statistical associations only, one cannot assess the causality or even try to do that. There is no proof that IA would have been already at level of 24.4 % among Taiwanese junior high school students prior to Covid-19 outbreak (even though this is quite plausible that the IA has increased due to Covid-19 outbreak). There is plenty of studies suggesting that the cause for the psychological symptoms (depression, neuroticism, self-esteem, alexithymia etc.) examined, like in this study are also due to socioeconomic factors and the family background, like age, gender, family social status, family income, marital status of parents etc. Some references below. None of these were taken into account in this study. It could well be that the students from families of high risk socioeconomic factors were the ones who started to use more internet after Covid-19 outbreak began. That would indicate that IA is more of a confounder for the true reasons of depression, neuroticism etc. It is impossible to assess the study as none of these, commonly recorded factors of socioeconomic nature, were not taken into account. The least, other variables should be adjusted for these to account for any biases. Surely and hopefully, some basic demographics (gender, area of residence etc.) were collected.

DOI: 10.1371/journal.pone.0137506

DOI: 10.1186/s12889-018-5990-8

DOI: 10.1016/j.jad.2020.08.026

DOI: 10.1016/S0140-6736(13)62116-9

DOI: 10.1016/S0140-6736(11)60871-4

Ans: Thank you for the suggestion. We agree: “There is no proof that IA would have been already at level of 24.4 % among Taiwanese junior high school students prior to Covid-19 outbreak (even though this is quite plausible that the IA has increased due to Covid-19 outbreak), and one cannot assess the causality or even try to do that”, and thus we elaborated in the Discussion section and limitation: “First, the research was based on a cross-sectional design, and thus we cannot eliminate the possibility that IA may already be present among the 24.4 % Taiwanese junior high school students (IA-group) prior to COVID-19 outbreak. Further studies should conduct follow-up studies in order to clarify whether the rise in prevalence was due to the COVID-19 outbreak.” (p8, lines 306-309). On the other hand, thank you for your detailed carefulness and suggestions. I have included some basic demographics in「2.2.1. The socio-demographic measures」(p3, lines 111-112), including gender (female=1 and male=2), age, and marital status of parents. Further analysis has also been conducted (p5, lines 182 to p6, lines 206). Furthermore, variables such as family social status, family income, and area of residence were not collected in this study, but we have discussed it in the limitations section, and made note that it should be taken into account in further studies. (p8, lines 316-319).

Major comments

  1. 30 update numbers to also give the most recent numbers (even though the study was during early outbreak) t-test is for continuous variables and their means, your variables have only about 4-6 classes. E.g. chi-square might be more suited here.

Ans: Thank you for the positive affirmation in our effort to cite the latest references! In addition to using t-test, we have also included a chi-square test in the revised manuscript (p4, lines 171-172 & p5, lines 182-192).

Study and sampling design has not been described at all. In line 94, a sentence saying that “stratified and cluster sampling” was done, is not enough. There should be detailed description on where, how, with whom the sampling was done, ethical statements etc. Was any demographics collected, hopefully so, as it would save this study.

Ans: Thank you for your detailed carefulness. I have added a more detailed description on where, how, with whom the sampling was done, ethical statements etc (p3, lines 93-109). Moreover, some demographics were collected, including gender, age, and marital status of parents (p3, lines 111-112).

If in Taiwan, senior high school students have IA of 17.4 %, the 24.4 % among junior high school students obtained in this study is not much higher.

Ans: Thank you for your suggestion. The research team will continue to pay a close attention on the IA prevalence of junior and high school students in Taiwan.

Any difference between schools? Were schools from similar socioeconomic areas?

Ans: The three junior high schools that participated in the study were municipality, each located in Taipei City, New Taipei City, and Taoyuan City, respectively, which have similar socioeconomic features and may also limit the generalizability of study results. We have also included this statement in the limitations section. (p8, lines 309-315).

You give no alternative for junior high school students what to do during Covid-19 lockdowns. If internet is forbidden, then what? Going out with friends is not an option. I would see this far more positively, or at least it cannot be concluded that the side effects were negative. It could also be that the use of IA helped those students with these mental health issues to cope better. This could be possibly analyzed with stratified methods.

Ans: Thank you for your suggestions. I also agree that during the COVID-19 lockdown, there is a limited number of activities available for junior high school students, and thus Internet use is the most convenient and efficient. Previous research have also pointed out that stay-at-home quarantines and mandates have escalated the consumption of digital entertainment (p1, lines 33-34). Thus, adequate use of the Internet becomes an even more important lesson for students to learn, and how not to develop into IA. A healthy form of Internet use is deem acceptable and positive.

Reviewer 2 Report

First of all. Thank the authors for their work. They present a solid study with a significant sample. However, I must suggest certain amendments in order to improve the text presented and clarify certain issues.

1.- The introduction is well explained and relevant. However, due to the large number of variables studied, I believe that this last paragraph should list each of them and briefly explain their relationship to the study. In some ways, it is necessary to contextualize and justify the choice of those variables.

2.- The section of materials and methods is extensive and well done. However, I raise a question, despite the large sample, only 3 schools participate. This should then be indicated as a limitation and explain whether they belong to the same region.

3.- The results section is well raised. Perhaps some explanation could be expanded so as not to leave all the information in understanding tables.

4.- The discussion is well linked to the theory presented in the introduction and is consistent with the findings provided by the research.

5.- In the conclusions, authors should add limitations and prospective study. For example, even though these measures are taken in an alarm state by COVID 19, it cannot be assured that they were different before the pandemic. In addition, it would be wise to compare your study with others conducted in other cultures and at different times.

Author Response

In response to the comments from reviewer 2:

First of all. Thank the authors for their work. They present a solid study with a significant sample. However, I must suggest certain amendments in order to improve the text presented and clarify certain issues.

Ans: Thank you for your positive affirmation and suggestion.

  1. The introduction is well explained and relevant. However, due to the large number of variables studied, I believe that this last paragraph should list each of them and briefly explain their relationship to the study. In some ways, it is necessary to contextualize and justify the choice of those variables.

Ans: This study applied the theory of triadic influence in attempt to understand the psychosocial risk factors for IA in adolescents during the COVID-19 outbreak, and focused on the intrapersonal and social stream of influence based on the quarantines of the COVID-19 pandemic and stay-at-home mandates. Thank you for your suggestions I have listed each variable in the last paragraph and briefly explained their relationship to the study (p2, lines 84-87).

  1. The section of materials and methods is extensive and well done. However, I raise a question, despite the large sample, only 3 schools participate. This should then be indicated as a limitation and explain whether they belong to the same region.

Ans: Thank you for your positive affirmation and suggestion. We have included the participation of only 3 schools as a limitation (p8, lines 309-313), and explained that they were all municipality with similar socioeconomic features, which may also limit the generalizability of study results. (p8, lines 313-315).

  1. The results section is well raised. Perhaps some explanation could be expanded so as not to leave all the information in understanding tables.

Ans: Thank you for your positive affirmation and suggestion. In order to avoid the issue of leaving all the information in understanding tables, we have elaborated more explanation in the results section to (p5, lines 182-190).

  1. The discussion is well linked to the theory presented in the introduction and is consistent with the findings provided by the research.

Ans: Thank you for your positive affirmation!

  1. In the conclusions, authors should add limitations and prospective study. For example, even though these measures are taken in an alarm state by COVID 19, it cannot be assured that they were different before the pandemic. In addition, it would be wise to compare your study with others conducted in other cultures and at different times.

Ans: Thank you for your suggestions. In the conclusions, I have added in the limitations that “Although results are found during the alarmed state by COVID-19, it cannot be assured that they were different before the pandemic, and further studies should compare the results with that of other cultures and at different times.” (p9, lines 339-342).

Reviewer 3 Report

This paper examines the determinants of internet addiction among high school students during the winter lockdown in Taiwan. The sample size is large and the information provided is adequate for a sound statistical modeling. The paper is concise, clear and well-written. The research questions posed are also sound and interesting. I only have two minor comments to the authors:

1) page 5, rows 189-197: you find that neuroticism, depression, actual social support adn self-esteem are not significant in the regression model when all covariates are controlled for. However, in a univariate regression model, each of these variables are statistically significant in explaining variations in the internet addiction index. I believe you should try to analyse more of this finding. These variables seems to be theoretically important determinants of the dependent. Why such findings arise? It is possible due to the high correlations observed in Table 2 between these variables. Could you reform the model appropriately (by combining for example two or more of these variables in a new indicator) so you can have some findings about these dimensions of the theoretical model as well? In addition, is any of these variables statistically significant if you control for each one separately in the basic regression model of Table 3? Perhaps this tests can give an indication of the confounding relationships between indicators that seem to be highly related.

2) page 4, rows 165-167: you state that higher scores in the family function index indicate healthier family function. However, in Table 3 you find a positive relationship with internet addiction. The positive sign indicates that students of families with healthier functioning have a higher probability to experience higher internet addiction scores in comparison with the remainder. In page 7, rows 275-287, you interpret the estimated effect in the opposite way. Perhaps I did not understand this point well, but I believe you should clarify better what this positive sign means.

Author Response

In response to the comments from reviewer 3:

This paper examines the determinants of internet addiction among high school students during the winter lockdown in Taiwan. The sample size is large and the information provided is adequate for a sound statistical modeling. The paper is concise, clear and well-written. The research questions posed are also sound and interesting. I only have two minor comments to the authors:

Ans: Thank you for your positive affirmation!

1) page 5, rows 189-197: you find that neuroticism, depression, actual social support and self-esteem are not significant in the regression model when all covariates are controlled for. However, in a univariate regression model, each of these variables are statistically significant in explaining variations in the internet addiction index. I believe you should try to analysis more of this finding. These variables seems to be theoretically important determinants of the dependent. Why such findings arise? It is possible due to the high correlations observed in Table 2 between these variables. Could you reform the model appropriately (by combining for example two or more of these variables in a new indicator) so you can have some findings about these dimensions of the theoretical model as well? In addition, is any of these variables statistically significant if you control for each one separately in the basic regression model of Table 3? Perhaps this tests can give an indication of the confounding relationships between indicators that seem to be highly related.

Ans: Thank you for your careful and precise suggestion! We reanalyzed the data with logistic regression analyses. We included the variables of impulsivity, virtual social support, age, subjective well-being, family function, and alexithymia in the analysis, and then we independently added actual social support (p=0.814), neuroticism (p=.804), depression (p=.315), and self-esteem (p=.559), all showing no significance in predicting IA. In addition, we also included the variables of impulsivity, virtual social support, age, subjective well-being, family function, and alexithymia in the analysis, and then we independently added neuroticism + depression (p=0.740), neuroticism + depression + self-esteem (p=0.852), neuroticism + depression + self-esteem + actual social support (p=0.799), all showed non-significance in predicting IA.

2) page 4, rows 165-167: you state that higher scores in the family function index indicate healthier family function. However, in Table 3 you find a positive relationship with internet addiction. The positive sign indicates that students of families with healthier functioning have a higher probability to experience higher internet addiction scores in comparison with the remainder. In page 7, rows 275-287, you interpret the estimated effect in the opposite way. Perhaps I did not understand this point well, but I believe you should clarify better what this positive sign means.

Ans: I carefully inspected our data again. In Table 3, this study discovered a “negative” relationship between family function and Internet addiction (Odd Ratios=0.982, and 95% confidence interval is between 0.970—0.994) (p6, lines 205-206). Thank you for your rigorous precision!